# Advances in the Synthesis and Biological Applications of Enoxacin-Based Compounds

**DOI:** 10.3390/biom14111419

**Published:** 2024-11-07

**Authors:** Garba Suleiman, Nabil El Brahmi, Gérald Guillaumet, Saïd El Kazzouli

**Affiliations:** 1Euromed Research Center, School of Engineering in Biomedical and Biotechnology, Euromed University of Fes (UEMF), Fez 30000, Morocco; s.garba@ueuromed.org (G.S.); n.elbrahmi@ueuromed.org (N.E.B.); 2Institut de Chimie Organique et Analytique, Université d’Orléans, UMR CNRS 7311, BP 6759, CEDEX 2, 45067 Orléans, France

**Keywords:** antibacterial, biological activity, anticancer, enoxacin, fluoroquinolone, naphthyridine, synthesis, quinolone

## Abstract

A comprehensive review of advances in the synthesis and biological applications of enoxacin (1, referred to as ENX)-based compounds is presented. ENX, a second-generation fluoroquinolone (FQ), is a prominent 1,8-naphthyridine containing compounds studied in medicinal chemistry. Quinolones, a class of synthetic antibiotics, are crucial building blocks for designing multi-biological libraries due to their inhibitory properties against DNA replication. Chemical modifications at positions 3 and 7 of the quinolone structure can transform antibacterial FQs into anticancer analogs. ENX and its derivatives have been examined for various therapeutic applications, including anticancer, antiviral, and potential treatment against COVID-19. Several synthetic methodologies have been devised for the efficient and versatile synthesis of ENX and its derivatives. This review emphasizes all-inclusive developments in the synthesis of ENX derivatives, focusing on modifications at C3 (carboxylic acid, Part A), C7 (piperazinyl, Part B), and other modifications (Parts A and B). The reactions considered were chosen based on their reproducibility, ease of execution, accessibility, and the availability of the methodology reported in the literature. This review provides valuable insights into the medicinal properties of these compounds, highlighting their potential as therapeutic agents in various fields.

## 1. Introduction

Quinolones, a class of synthetic antibiotics, are widely recognized as crucial building blocks for designing multi-biological libraries [1,2]. Their inhibitory properties against DNA replication make them effective against various pathogens, including mycoplasma, bacteria, and protozoa [3,4,5]. These synthetic antibacterial drugs belong to the broader class of fluoroquinolones (FQs) and act by targeting DNA gyrase, topoisomerase enzymes, and topoisomerase IV, which are involved in DNA replication and repair processes in bacteria [6,7,8,9,10,11,12,13].

The discovery of nalidixic acid in 1962 marked the beginning of the use of quinolone derivatives as antibacterial agents worldwide [13,14]. The subsequent development of FQs in the 1970s and the 1980s significantly expanded their coverage [15,16]. FQs exhibit diverse biological activities, including against infectious diseases such as malaria and parasitic, bacterial, and fungal diseases [3,17,18,19], as well as viral infections such as hepatitis, human immunodeficiency virus (HIV), and herpes [20]. They are highly effective against Gram-negative Pseudomonas infections and have been employed in treating pneumonia and intra-abdominal infections [21]. Additionally, they show promise in treating autoimmune diseases, organ transplantation, and rheumatoid arthritis with low toxicity [2,22,23,24]. FQs can impede tumor growth by inducing damage to type II human DNA topoisomerases, similar to specific chemotherapy drugs such as etoposide [25,26], making them noteworthy agents in infectious disease management and potential adjuncts in certain cancer treatment strategies. 

The critical structural attributes of quinolones have been identified, with 4-oxo-quinolone-3-carboxylic acid being a significant substructure in numerous quinolone derivatives with outstanding biological activities [27,28]. Chemical modifications at position 7 transform antibacterial FQs into anticancer analogs, while the carboxylic group at position 3 plays a vital role in enzyme binding and functional group transformation, enhancing anticancer potential [27,29,30]. FQs such as levofloxacin and moxifloxacin are designated by the WHO as second-line drugs for treating tuberculosis due to their broad and potent spectrum of activities as well as oral administration [31,32,33]. The versatility of quinolones and FQs makes them valuable tools in medicinal research and therapeutic applications across different disciplines.

FQs with a 1,8-naphthyridine core are a specific subset of the fluoroquinolone class, where the quinolone nucleus is replaced by a naphthyridine structure. In the case of FQs with a 1,8-naphthyridine core, the compounds primarily differ at two key positions, N1 and C7, with modifications often occurring at C3 and C7. Figure 1 depicts the 1,8-naphthyridine core, clearly labeling N1 through N8 to emphasize these distinctions within the structure. To illustrate, enoxacin (1, referred to as ENX) is known for having a piperazinyl group at C7 and an ethyl group at N1. In contrast, gemifloxacin, while also featuring the 1,8-naphthyridine core, has an aminopyrrolidinyl group at C7 and a cyclopropyl group at N1. Other FQs with this core typically have a different group at the C7 and N1 positions, as illustrated in Figure 2.

In 1980, ENX, a 1,8-naphthyridine derivative of nalidixic acid, was discovered [34]. Although six distinct isomeric forms of naphthyridine exist, 1,8-naphthyridine derivatives have been extensively researched [35,36,37]. This unique skeleton has led to various bioactive compounds derived from natural sources, demonstrating significant biological applications [38,39,40]. ENX, a fluorinated antibacterial drug, and voreloxin, a non-fluorinated potential anticancer agent, are prominent 1,8-naphthyridines studied in medicinal chemistry [26,41]. Other important 1,8-naphthyridine-containing molecules with demonstrated biological activity include nalidixic acid, trovafloxacin, tosufloxacin, voreloxin, and gemifloxacin (Figure 2).

ENX, a second-generation fluoroquinolone, is known for its wide-spectrum antibacterial activity against both Gram-positive and Gram-negative bacteria [42,43,44]. Structurally, ENX comprises two fused six-membered rings with a 1,8-naphthyridine core as the parental structure (Figure 3) [45,46]. This drug is often well-tolerated and has a low frequency of side effects. It is typically delivered orally in the form of tablets. However, due to the development of resistance by many strains of bacteria, including *Escherichia coli* (*E. coli*) and *Pseudomonas aeruginosa* (*P. aeruginosa*), it is no longer considered a first-line treatment for bacterial infections [47]. Over the past few decades, scientists have examined the potential usage of ENX and its derivatives for several therapeutic applications [48,49,50]. In vitro tests have revealed that ENX exerts significant cytotoxicity in human cancer cells [48,51]. Moreover, it has also been reported to enhance the anticancer effects of other chemotherapeutic medications, including paclitaxel [51,52,53]. In addition, ENX possesses antiviral properties, making it effective against many different infections, including HIV and hepatitis C virus (HCV) [48,52].

A recent study on repositioning FQs demonstrated the potential of repurposing ENX for its use as a potential treatment against COVID-19 (SARS-CoV-2) [54,55,56]. Although there are many motives for reviewing the chemical synthesis of ENX and its derivatives, some of the critical reasons are selectivity [57], repositionability [51], oral bioavailability [58], a better safety profile, pro-oxidative activity, and regulation of microRNA biogenesis [59]. ENX ‘s unique microRNA-interfering activity sets it apart from other FQs and topoisomerase II drugs [45].

Several synthetic methodologies have been devised and implemented and are known for their efficiency, versatility, and convenience [1,60,61]. However, no exhaustive review has exclusively presented the synthesis of ENX and its derivatives based on the current literature and understanding [62]. In this review, we highlight key developments in the synthesis of 4-quinolone-3-carboxylic acid derivatives with a 1,8-naphthyridine core, specifically focusing on ENX, and discuss its medicinal properties where relevant. Recent publications have discussed the expanded therapeutic potential of diverse heterocyclic molecules beyond their conventional applications [63,64,65,66].

The present analysis is structured into three distinct segments: part A focuses on the modification of the carboxylic acid at the C3 position, part B addresses the modification of the piperazinyl group at the C7 position, and the section on other modifications explores combined modifications involving both parts A and B. The reactions considered in this review were chosen based on their capacity for reproducibility, relative ease of execution, accessibility, and the availability of the methodology as reported in the literature. Below is the structural representation of ENX with labeled atom positions comprising the C3-carboxylic part, the C7-piperazinyl part, and the fluoroquinolone core (Figure 3).

## 2. Modifications of ENX-Based Compounds

### 2.1. C3 Modification of ENX (Part A)

In 2009, You and colleagues [67] designed and synthesized a novel series of quinolone and naphthyridine derivatives as potential topoisomerase I inhibitors by modifying the scaffold in three steps. The first step involved condensation of ENX with **2** in polyphosphoric acid (PPA) at 170–250 °C to obtain **3a**–**c** or **4** (Table 1). In the subsequent step, intermediate **3a**–**c** was nitrated in a mixture of concentrated sulfuric acid (H_2_SO_4_) and nitric acid (HNO_3_) in an approximately equal ratio at 5 °C, followed by heating at 40–45 °C for 1–2 h, yielding **5a**–**c**. In the final step, the nitro-containing compound **5c** was subjected to hydrogenation over Pd/C in 1 N hydrochloric acid (HCl) solution to produce **6** (Figure 1). All derivatives containing three kinds of heterocycles, benzoxazole, benzimidazole, and benzothiazole, at the C3 position were screened in vitro for their antiproliferative effects against oral epidermal carcinoma (KB), ovarian carcinoma (A270), and hepatocellular carcinoma (Bel-7402) cells using a 1-*N*-methyl-5-thiotetrazole (MTT)-based assay (Table 1). In summary, the 3-benzothiazolenaphthyridine skeleton **3c** showed the highest antiproliferative activity (IC_50_ = 2.4–2.7 μM) against three tumor cell lines. Conversely, nitro-containing 3-benzoxazolenaphthyridine scaffold **5b** displayed even better cytotoxic activity (IC_50_ = 31.8–3.0 μM). Surprisingly, reducing the nitro group in **5a** to **6** resulted in significantly diminished cytotoxicity. This reinforces the hypothesis that an electron-withdrawing group is essential for cytotoxic activity.

A few years later, Yang and coworkers [68] synthesized 1,8-naphthyridin-3-yl-1*H*-benzo-6-carbonitrile derivatives of ENX by replacing the carboxyl group at C3 with a 2,3-dihydro-1*H*-benzimidazole-5-carbonitrile system in a single step, employing the same procedure as described in Figure 1 [67]. The target compound was realized by condensing ENX with **7** at 170–250 °C in PPA to yield product **8** (Figure 2). Their studies were primarily centered around investigating the potential molecular mechanism by which it exhibits its antitumor activity against non-small-cell lung cancer (NSCLC). The results revealed that compound **8** exhibited significantly stronger inhibitory effects against NSCLC compared to its leading compound ENX, both in cultured cells and in a xenograft mice model. It also increases reactive oxygen species (ROS) generation and DNA damage response (DDR) in a dose-dependent manner. The ROS scavenger *N*-acetyl-cysteine (NAC) reduced DDR and apoptosis triggered by **8**, confirming that its antitumor actions are due to oxidative stress. Thus, **8** promotes oxidative stress and cell death by activating the mitochondrial and endoplasmic reticulum (ER) stress pathways [68].

In a study conducted by Arayne and colleagues [69], the synthesis of carboxy-substituted ENX analogs as antibacterial agents was documented. This synthesis involved the amidation of the 3-carboxylic acid group of ENX using aromatic amines (RNH_2_) and phenyl hydrazine. Initially, an ENX ester, **9**, was prepared in methanol with a catalytic amount of H_2_SO_4_ at reflux for 7–8 h. The resulting intermediate **9** was further reacted with different aromatic amines, as well as phenyl hydrazine, under reflux for 2–3 h, yielding the desired carboxamides **10a**–**d** and the carbohydrazide **11** in moderate to good yields (Figure 3). Compounds **10a**–**d** and **11** were tested against various bacteria, revealing remarkably improved antimicrobial effectiveness against Gram-negative strains. Furthermore, their potential to influence the immune response was assessed in a separate study [70]. To evaluate their immunomodulatory activity, the impact on the oxidative burst activity of phagocytes in whole blood, as well as macrophages and neutrophils, was investigated. Among the synthesized derivatives, compounds **10c** and **10d** exhibited the highest level of inhibition in whole blood (IC_50_ = 2.6 and 1.4 µg/mL), macrophages (IC_50_ = 3.2 and 1.4 µg/mL), and isolated neutrophils (IC_50_ = 0.8 and 1.4 µg/mL), respectively (Table 2).

### 2.2. C7 Modification of ENX (Part B)

According to the literature, C7 piperazinyl quinolone modifications are effective not only against Gram-positive and Gram-negative pathogens [71] but also have numerous biological applications against cancer [72,73], inflammation [28], osteoclasts [74], viral infections [75], and other diseases [76,77]. As prospective osteo-adsorptive drugs, Herczegh and coworkers [78] developed a series of bisphosphonate FQ derivatives. The piperazinyl group of ENX was transformed with tetraethyl ethene-1,1-diylbis(phosphonate) **12**. In the first step, ENX was combined with **12** in the presence of triethylamine (Et_3_N) in dichloromethane (DCM), under stirring at room temperature (rt), for 3 h. Afterwards, an aqueous work-up and recrystallization from toluene produced the bis-(diethoxy-phosphoryl)-ethyl ester **13**. The ester was then hydrolyzed with bromotrimethylsilane (CH_3_)_3_SiBr in DCM at rt for 72 h, yielding **14** as a hydrobromide salt. Treatment of the salt with water (H_2_O) at rt for 6 h, followed by agitation in DCM and subsequent ether washing, resulted in an average yield of the desired compound, bis-phosphonic-ENX derivative **14** (Figure 4).

In another study, Vracar and colleagues [79] discovered that ENX and bis-phosphonic-ENX, **14**, have been found to induce the release of extracellular vesicles from 4T1 murine breast cancer cells, which possess inhibitory effects on osteoclastogenesis. Surprisingly, adding a bisphosphonate moiety boosted bone binding affinity. Moreover, bis-phosphonic-ENX, similar to ENX, displayed inhibitory effects on the binding of V-ATPase to microfilaments, as well as on bone resorption in vitro. In summary, bis-phosphonic-ENX offers multiple benefits beyond preventing bone mineral loss. It not only modifies the composition of bone glycoproteins, making them more resistant to fractures, but also completely suppresses osteoclast differentiation. Both ENX and bis-phosphonic-ENX demonstrate similar potency, with IC_50_ values around 10 µM, indicating their strong inhibitory effects on osteoclasts.

Darekhordi and colleagues [80] reported the synthesis of the medicinally important ENX derivative **16** under moderate conditions in a single-step approach. The synthesis involved reacting ENX with **15** using potassium carbonate (K_2_CO_3_) in dimethylformamide (DMF) at reflux for 24 h, yielding **16** in a reasonable yield (Figure 5). In addition, the antibacterial efficacy of the synthesized conjugate was tested via the agar diffusion method and exhibited concentration-dependent improved activity against *E. coli*, *Klebsiella pneumoniae* (*K. pneumoniae*), and *Staphylococcus aureus* (*S. aureus*).

In their study, Xiao and colleagues [81] described the synthesis of FQ–flavonoid hybrids using a well-designed pharmacophore system, aiming to develop a multi-target bacterial topoisomerase inhibitor with potential as an efflux pump inhibitor. The synthesis involved the reaction of ENX with different flavonoids (**17)**, such as apigenin and naringenin, while including an ethylene linker in the process (Figure 6). In the initial step, **17a**–**c** was *o*-selectively alkylated with 1,2-dibromoethane in the presence of K_2_CO_3_ in DMSO at 70 °C for 15 h, yielding compounds **18a**–**c**. Then, compounds **18a**–**c** were reacted with ENX in DMSO in the presence of DMAP at 60 °C for 40–50 h, yielding FQ–flavonoid hybrids **19a**–**c** in reasonable yields (55–75%). The antibacterial efficacy of the hybrids was tested against different microorganisms, including Tetracycline-resistant *Bacillus subtilis* ATCC 6633 (*B. subtilis*), amphotericin B-resistant *Candida albicans* (*C. albicans*), multiple drug-resistant *E. coli* ATCC 35218, and methicillin-resistant *S. aureus* ATCC 25923. Some of these compounds displayed impressive antibacterial properties, particularly against drug-resistant strains. Remarkably, derivative **19a** exhibited outstanding activity against *B. subtilis* and *C. albicans* with minimum inhibitory concentrations (MICs) of 0.45 µg/mL and 2.60 µg/mL in comparison to the standard drug ciprofloxacin (CPX), which had MIC values of 2.70 µg/mL and 32.4 µg/mL for the respective microorganisms (Table 3).

A methylene-bridged nitrofuran *N*-substituted piperazinylquinolone was designed and synthesized by Emami and colleagues [82]. ENX mixed with 2-(bromomethyl)-5-nitrofuran **20** in DMF in the presence of sodium hydrogen carbonate (NaHCO_3_) as a base at rt for 120 h resulted in the formation of the desired compound **21** (Figure 7) at a good yield (81%). The antibacterial assessment demonstrated that the efficacy of 7-piperazinylquinolones with (5-nitrofuran-2-yl) derivatives against diverse bacterial strains is contingent upon the nature of the substituents located at the N1 and C7 sites. Overall, the compound displayed noteworthy antibacterial efficacy against *Staphylococci* in a manner that was dependent on their concentration. Compound **21** showed the best inhibitory activity against *S. aureus* with a MIC of 0.39 μg/mL.

In another report [83], four novel ENX derivatives were synthesized by introducing 2-(5-chlorothiophen-2-yl)ethyl into the piperazine ring. The synthesis was performed by reacting ENX with intermediates **22a**–**d** in DMF at rt, employing NaHCO_3_ and yielding **23a**–**d** in 62–73% yields (Figure 8). The introduction of 2-(5-chlorothiophen-2-yl)ethyl into the piperazine ring of ENX resulted in enhanced cytotoxicity against various cancer cell lines compared to the unmodified ENX [84]. Compound **23** exhibited varying modifications to the ethyl spacer structures. Regarding their cytotoxicity against cancer cell lines, including melanoma (SKMEL-3), breast (MCF-7), epidermoid (A431), bladder (EJ), colon (SW480), and KB cell lines, compounds **23b** and **23c** demonstrated the most significant impact. Specifically, **23b** displayed an IC_50_ range of 3 to 10 μM, while **23c** showed an IC_50_ range of 3 to 20 μM (Table 4). On the other hand, **23d** exhibited IC_50_ values of 2 to 14 μM for melanoma, epidermoid, cervical, and bladder cell lines. In summary, incorporating the 2-(5-chlorothio-phen-2-yl)ethyl group into the piperazinyl portion of ENX enhanced its cytotoxic properties compared to the parent ENX, although the extent of improvement depended on the structure of the spacer. By introducing an additional functionality, the antitumor effectiveness rose considerably (Table 4).

Synthesis and pre-formulation studies were conducted on a pharmacologically inactive precursor of ENX, resulting in the synthesis of **24** [85]. The synthesis involved reacting ENX with formaldehyde (CH_2_O) in a solution of dichloromethane and methanol mixed in an equal ratio at rt for 3 h. The resulting compound was obtained in 89% yield (Figure 9). The antimicrobial effectiveness of the prodrug was evaluated in comparison to ENX using the agar diffusion method, specifically targeting *E. coli*, *P. aureginosa*, and *S. aureus*. The most noteworthy outcome was observed against *E. coli*, where the MIC was determined to be 0.2 μg/mL.

*N*-substituted piperazinyl quinolone **26** was synthesized and examined for in vitro antibacterial activity against various strains of bacteria [86,87]. Through the reaction of ENX with **25** and NaHCO_3_ in DMF at 85–90 °C for 12 h, **26** was obtained in satisfactory yield (Figure 10). The antibacterial evaluation demonstrated that **26** exhibited potent and superior activity against the tested Gram-positive bacteria compared to reference FQs such as ENX. Compound **26** exhibited the highest activity against *B. subtilis*, with a MIC value of 0.008 μg/mL, surpassing the ENX value of 0.125 μg/mL.

Foroumadi et al. [88] reported a series of *N*-substituted piperazinyl quinolones using thiadiazole derivatives **27** with ENX and NaHCO_3_ in DMF at 85–90 °C for 12 h (Figure 11). This method successfully synthesized bioactive derivatives of *N*-[5-(chlorobenzylthio)-1,3,4-thiadiazol-2-yl] piperazinyl quinolones **28a**–**d** in moderate yields (62–67%). To evaluate the efficacy of the synthesized compounds, the agar dilution method was employed against a panel of bacteria including *S. aureus*, *Staphylococcus epidermidis* (*S. epidermidis*), *B. subtilis*, *E. coli*, *K. pneumoniae*, and *P. aeruginosa.* The results indicate that the obtained derivatives exhibited moderate antibacterial activity against the tested microorganisms (Table 5).

In a similar study, a variety of ENX-substituted derivatives **30a**–**g** were synthesized and tested for antibacterial activity in vitro by combining the ENX with appropriate intermediates **29a**–**g** [89]. The target derivatives were obtained through the *N*-alkylation of ENX with properly substituted intermediates **29a**–**g** by employing NaHCO_3_ as a base in DMF as an appropriate solvent in good yields (76–79%) (Figure 12). The in vitro antibacterial activity of **30a**–**g** against various bacterial strains revealed that compounds **30a**–**c** and **30g** demonstrate antibacterial activity similar to ENX against certain bacterial strains, particularly Gram-positive bacteria such as Staphylococci and Gram-negative bacteria such as *E. coli* and *Enterobacter cloacae* (*E. cloacae*). However, none of the derivatives consistently outperformed ENX across all the tested strains (Table 6).

Foroumadi et al. [90] described the synthesis and antibacterial activity evaluation of piperazinyl-substituted ENX analogs **32a**–**d.** The synthesis involved reacting **31** with ENX using NaHCO_3_ in DMF at rt, resulting in the generation of ENX analogs **32a**–**d** in 45–72% yields (Figure 13). The synthesized derivatives were evaluated against a variety of bacterial strains. All the tested derivatives show appreciable antibacterial activity against *B. subtilis*, with inhibitory concentrations ranging from 1.56 to 6.25 μg/mL. Although **32b** has consistently shown moderate activity across the tested strains, none of the compounds **32a**–**d** demonstrated potent antibacterial effects that were comparable to the reference drug ENX (Table 7).

The same group [91] synthesized ENX furan-containing analogs from the furan-based intermediate **33**. *N*-[2-(furan-3-yl)-2-oxoethyl] or *N*-[2-(furan-3-yl)-2-oxyiminoethyl] **34a**–**d** was produced by treating ENX with **33** in the presence of NaHCO_3_ at rt in moderate yields (41–59%) (Figure 14). Evaluation of **34** against various bacterial strains revealed that **34a**–**c** exhibit comparable antibacterial activity to ciprofloxacin (CPX) against *S. aureus*, methicillin-resistant *S. aureus* (MRSA I and II), *S. epidermidis*, and *B. subtilis*. Specifically, compound **34a** has a MIC range of 0.39 to 0.78 μg/mL against these strains, which is similar to the MIC range of 0.19 to 0.39 μg/mL observed for CPX. Compound **34b** demonstrates a potency of 0.39 μM against *S. aureus*, MRSA, and *S. epidermidis*, closely matching the efficacy of CPX. Likewise, compound **34c** shows a MIC of 0.78 μg/mL against the same strains, again aligning with the antibacterial potency of CPX (Table 8).

Emami et al. [92] reported the synthesis and antibacterial evaluation of ENX coumarin-derived analogs **36a**–**d**. The synthesis of the hybrids required the reaction of ENX with coumarin-based precursors **35** (Figure 15). This reaction took place in DMF in the presence of NaHCO_3_ at rt for 6–72 h, resulting in the desired analogs **36a**–**d** in moderate to excellent yields (57–91%). The antimicrobial efficacy of **36a**–**d** was assessed using the agar diffusion method. Compound **36a** exhibits the most potent antibacterial activity across all tested bacteria, including *S. aureus*, MRSA I, MRSA II, *S. epidermidis*, *B. subtilis*, *E. coli*, and *K. pneumoniae*, with MIC values ranging from 0.049 to 3.13 μg/mL. Notably, **36a** shows comparable or superior activity to the reference compound ENX against *S. aureus*, MRSA I, MRSA II, *S. epidermidis*, *B. subtilis*, and *E. coli.* Compound **36b** also demonstrates significant antibacterial activity, with MIC values between 0.39 μg/mL and 12.5 μg/mL. However, **36b** is generally less potent compared to ENX. On the other hand, compounds **36c** and **36d** exhibit weaker antibacterial potency compared to both **36a** and **36b**, with MIC values that are generally higher than those of ENX (Table 9).

Shafiee et al. [93] documented the synthesis and antibacterial activity of naphthyl-containing ENX analogs **38a**–**d**. The desired compounds were successfully synthesized using a versatile and efficient synthetic pathway (Figure 16). This approach involved reacting ENX with **37** in the presence of NaHCO_3_ in DMF at rt for 72 h. The resulting products were obtained in good yield (51–83%). The antibacterial evaluation of these derivatives demonstrated promising activity against the tested analogs. Compound **38a** displays comparable or superior antibacterial activity to ENX across all tested strains, with IC_50_ values ranging from 0.049 to 0.780 μg/mL. Similarly, **38b** shows superior activity compared to ENX, particularly against *B. subtilis* and *E. coli*, with IC_50_ values of 0.190 and 0.390 μg/mL, respectively. In contrast, compounds **38c** and **38d** generally exhibit weaker antibacterial activity compared to **38a** and **38b**, as well as the reference compound ENX (Table 10).

Ahmed and colleagues [94] conducted a groundbreaking study where they skillfully synthesized and screened new alternative molecules of ENX derivatives as potential antibacterial and antibiofilm agents (Figure 17). ENX was acylated with acid chlorides **39** using Et_3_N as a base in refluxing tetrahydrofuran (THF). The desired products **40a**–**e** were obtained with a moderate yield (49–64%). Evaluation of the antimicrobial potential of **40** against a panel of pathogens via the micro-broth dilution method revealed that all the synthesized derivatives were found to be active at low concentrations against MRSA, *K. pneumoniae*, *and Proteus mirabilis* (*P. mirabilis*) with MIC values in the range of 12.5 to 25 μg/mL compared to the parent molecule, ENX. Specifically, compounds **40b**, **40c**, and **40e** inhibited the growth of MRSA at a 1 μg/mL concentration better than the parent drug ENX. The antibiofilm inhibitory properties of the synthesized derivatives revealed that **40b**, **40c**, and **40e** inhibited MRSA biofilm formation in the concentration range of 0.5 to 1 μg/mL (Table 11).

Wang and coworkers [95] generated a library of 3-arylfuran-2(*5H*)-one-fluoroquinolone hybrids **46a**–**e**. Initially, substituted phenylacetic acids **41a**–**e** were converted to sodium phenylacetates **42a**–**e** in a dilute NaOH solution. Subsequent treatment of the intermediate salt with ethyl bromoacetate in DMSO at rt for 4 h resulted in the formation of phenylacetic acid ethyl esters **43a**–**e** in excellent yields (90–95%). Cyclization of **43a**–**e** was accomplished using sodium hydride (NaH) in THF at 0 °C to rt, leading to the formation of 4-hydroxy-3-phenylfuran-2(*5H*)-ones **44a**–**e.** The introduction of an ethyl linker was achieved by dissolving **44a**–**e** in acetone and adding 1,2-dibromoethane and Et_3_N, followed by refluxing the mixture for 3–5 h, resulting in the formation of compounds **45a**–**e** in good yields. Finally, the target products **46a**–**e** were realized in moderate yields by combining ENX with **45a**–**e** in the presence of KI and DMAP in DMSO at 60 °C for 72 h (Figure 18). The conjugated compounds were evaluated against a range of bacteria including tetracycline-resistant *B. subtilis*, *E. coli*, and *S. aureus*. Many of these analogs displayed antibacterial activity that was akin to the reference drug, CPX. Specifically, **46b** exhibited superior antibacterial efficacy across all the tested bacteria, with MIC_50_ values ranging from 1.6 to 2.6 μg/mL, which were significantly better than CPX with MIC_50_ values between 2.7 and 6.82 μg/mL (Table 12).

Shaheen et al. [96] developed and produced a series of novel FQs that exhibit strong inhibitory effects on *α*-glucosidase (Figure 19). The analogs were prepared by subjecting ENX to reflux conditions with various substituted benzyl chlorides **47a**–**g** in anhydrous acetone, in the presence of K_2_CO_3_, for 4–8 h. This process resulted in the desired monosubstituted compounds **48a**–**g** with satisfactory yields. The synthesized derivatives were then subjected to in vitro screening for α-glucosidase inhibition, along with in silico docking studies. The analogs **48a**–**g** demonstrated strong α-glucosidase inhibitory activity ranging from 48.7 to 74.5 μM, in comparison to the IC_50_ value of 425.6 μM observed for the reference *α*-glucosidase standard inhibitor drug, 1-deoxynojirimycin (Table 13). Docking studies of **48a**–**g** reveal that the molecular interactions of mono-benzylated derivatives align well with their inhibitory activity. These compounds were observed to form polar contacts with the active site of proteins, mainly involving residues such as Glu771, Asp392, Trp391, and Arg428.

### 2.3. Other Modifications (Parts A and B)

This category encompasses modifications performed on both the **C3** and **C7** sites of the 1,8-naphthyridine core of ENX derivatives.

In the same report, Shaheen and colleagues [96] developed and produced novel di-substituted benzyl FQ derivatives with excellent α-glucosidase inhibitory effects (Figure 20). The analogs were prepared as demonstrated in Figure 19. However, in this case, the ENX was refluxed with various substituted benzyl bromides **49a**–**c** in the presence of K_2_CO_3_ for 4–8 h, resulting in the formation of disubstituted derivatives **50a**–**c**. The in vitro α-glucosidase inhibition screening showed that compound **50a** had the highest potency among all tested analogs, with an IC₅₀ value of 45.8 μM. Other analogs in this series, **50b** and **50c**, also exhibited notable inhibitory activity, with IC₅₀ values of 67.8 μM and 59.8 μM, respectively. These values are significantly lower than the IC₅₀ of 425.6 μM for the reference α-glucosidase inhibitor. Interestingly, **50a** is not only more potent than the reference drug but also surpasses the parent compound, ENX, which has an IC₅₀ of 58.9 μM. Specifically, **50a** is about 9.3-fold more potent than the reference drug, stressing its strong potential as a lead candidate for further development. Docking studies of compounds **50a**–**c** indicate that their molecular interactions are consistent with their observed inhibitory activity. These studies show that the di-benzylated derivatives form polar contacts with the active site of the enzyme, primarily interacting with residues such as Gly566, Glu771, Trp391, Asp508, Arg428, and Asp392 (Table 14).

## 3. Future Perspectives

The recent developments discussed in this review shed light on the synthesis of 4-quinolone-3-carboxylic acid derivatives, with a particular focus on scaffolds containing a 1,8-naphthyridine core reminiscent of ENX. These advancements pave the way for future exploration and innovation in this field. One promising avenue for future research is the further exploration of C3 modifications, as they have shown potential for generating diverse analogs with improved medicinal properties. By employing strategic modifications at the C3 position, researchers can fine-tune the pharmacological profile of these compounds, enhancing their efficacy and reducing potential side effects. Additionally, the C7 modification segment warrants further investigation, as it offers opportunities to optimize the physicochemical properties and biological activities of 4-quinolone-3-carboxylic acid derivatives. By carefully manipulating the C7 position, researchers can potentially enhance the bioavailability, target specificity, and overall therapeutic potential of these compounds. Lastly, the approach that combines modifications from both parts A and B (other modifications) presents a promising direction for the design and synthesis of novel enoxacin derivatives with diverse pharmacological applications. Within this framework, researchers can explore a wide range of structural modifications in order to produce analogs with specialized features and unique biological activities. Overall, these prospects for the future emphasize the intriguing possibility for further breakthroughs in the synthesis and research of 4-quinolone-3-carboxylic acid derivatives.

## 4. Conclusions

In conclusion, this review provides a comprehensive analysis of developments in the synthesis of 4-quinolone-3-carboxylic acid derivatives, focusing on scaffolds containing a 1,8-naphthyridine core akin to ENX. The reviewed literature showcases various modifications at the C3 and C7 positions, as well as their combination, demonstrating their impact on the structural diversity, medicinal properties, and potential pharmacological applications of these compounds. The chosen reactions were selected based on their reproducibility, ease of execution, and the accessibility of the described methodologies. Researchers seeking to design and synthesize novel ENX derivatives with diverse pharmacological activities will find the insights presented in this review both valuable and insightful. This comprehensive analysis sets the stage for future investigations, where researchers can explore the untapped potential of 4-quinolone-3-carboxylic acids, specifically ENX derivatives, thereby opening new avenues for drug discovery and therapeutic interventions.

## Data Availability

Not applicable.

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
