# Peer review of "Advances in the Synthesis and Biological Applications of Enoxacin-Based Compounds"

_biomolecules, 2024, doi:10.3390/biom14111419_

Round 1
Reviewer 1 Report
Comments and Suggestions for Authors
The manuscript by S. El Kazzouli et al. entitled “Advances in the synthesis and biological applications of enoxacin based compounds” comprehensively reviews the family of title compounds. I find the topic interesting. I found some minor issues in the text, and therefore I suggest a minor revision before publication. My comments are listed below.
1) There is a confusion with compound numbering. If I am not mistaking there is no compound 1, compound 23. N-hydroxymethylenoxacin is numbered as compound 25 in the text (page 9) and on on Scheme 9 as 24.
2) Are titles 2.1 and 2.2 correct? The title of section 2.1 is “C3 modification of ENX (part A) “, but on Figure 3 part attached to C3 atom is denoted as part B. Similar situation is with 2.2 section. It should be checked.
3) It is hard to figure out what are substituents R in structures given in schemes without looking at tables. Is there an option to incorporate info on substituents into Schemes?
Author Response
Comments and Suggestions for Authors
The manuscript by S. El Kazzouli et al. entitled “Advances in the synthesis and biological applications of enoxacin based compounds” comprehensively reviews the family of title compounds. I find the topic interesting. I found some minor issues in the text, and therefore I suggest a minor revision before publication. My comments are listed below.
Thank you very much for your valuable feedback. We appreciate your positive assessment of our paper's quality, design, and structure with the scope of your journal.
- There is a confusion with compound numbering. If I am not mistaking there is no compound 1, compound 23. N-hydroxymethylenoxacin is numbered as compound 25 in the text (page 9) and on on Scheme 9 as 24.
Response 1:
Thank you for this insightful comment. Regarding the numbering, compound 1 represents enoxacin, denoted as ENX throughout the text as first appeared on page 2, line 64. This has also been updated in Scheme 1 and can be found on page 4, line 142-143. For compound 23, there was an oversight as we unintentionally skipped one number, leading to confusion between compounds 23, 24, and 25. However, the issues have been resolved from Scheme 3 through 9, re-numbering the compounds in between the two schemes from 11 through 24 (Scheme 9, page 10). Additionally, compound 24 has been updated in the main text and can be found on line 272, page 9.
- Are titles 2.1 and 2.2 correct? The title of section 2.1 is “C3 modification of ENX (part A) “, but on Figure 3 part attached to C3 atom is denoted as part B. Similar situation is with 2.2 section. It should be checked.
Response 2:
Thank you for this comment. Yes, the two titles (2.1 and 2.2) are correct, however, there was an oversight in naming part A and B on Figure 3 that led to interchanging part A for part B. That oversight has been updated and can be found in page 4, Figure 3.
3) It is hard to figure out what are substituents R in structures given in schemes without looking at tables. Is there an option to incorporate info on substituents into Schemes?
Response 3:
We appreciate your observation regarding the description of the R substituents given in our schemes. In response to that, we have updated many of the schemes (1, 8, 14, 15, 16) for easy comprehension. However, some of the reasons we included tables where necessary are to make the schemes very presentable and provide the detail information like the structures of substituents with their corresponding activity values.

Reviewer 2 Report
Comments and Suggestions for Authors The review manuscript “Advances in the synthesis and biological applications of enoxacin based compounds” is devoted to the C3 and C7 synthetic modifications of enoxacin, and biological activities of the resulting products. The review is organized as a very long historical introduction (66 references), and a review of chemical reactions (30 references). The present review overlaps very strongly with an earlier review paper by other authors (ref. 62, Zahoor, et al., 2017). The present review, indeed, is just a part of ref. 62 with a very small additional literature reviewed, which can be assumed to be a plagiarism. That’s why, in my opinion, this is a clear reject.Author Response
Reviewer 2
Comments and Suggestions for Authors
The review manuscript “Advances in the synthesis and biological applications of enoxacin based compounds” is devoted to the C3 and C7 synthetic modifications of enoxacin, and biological activities of the resulting products. The review is organized as a very long historical introduction (66 references), and a review of chemical reactions (30 references). The present review overlaps very strongly with an earlier review paper by other authors (ref. 62, Zahoor, et al., 2017). The present review, indeed, is just a part of ref. 62 with a very small additional literature reviewed, which can be assumed to be a plagiarism. That’s why, in my opinion, this is a clear reject.
Response:
Thank you for your careful review and for bringing to our attention the potential overlapping phrases in our manuscript. We appreciate your efforts in ensuring the originality of the content. While some scientific phrases may appear similar to those in the referenced work, this does not indicate an actual overlap between the two studies.
We have taken your comments seriously and made significant revisions on every part of our work to address your concerns by revising all our texts and highlighting the improved version. However, it is important to note that the referenced work primarily focused on summarizing synthetic pathways and rarely discussed the potential biological activity of the modified structures. In contrast, we presented an all-inclusive modification of enoxacin emphasizing the synthetic pathway, including key details such as temperature and reaction time for reproducibility and ease of execution. Additionally, we have provided detailed structures of the reported derivatives, along with their biological targets and corresponding activity values. Where applicable, the SAR studies have also been incorporated. Moreover, we have included summary tables where necessary to facilitate a clearer understanding of the impact of substituents across each series presented in our work.
Perhaps a potential limitation of our study is that we only included papers featuring biologically active synthesized enoxacin analogs that are accessible and reproducible. It is also worth noting that all the research papers considered in our work are fully derived from their original source and properly cited. Considering these distinctions, we believe our submission is original in both its design and structure.
Reviewer 3 Report
Comments and Suggestions for Authors
Comments:
The review manuscript titled "Advances in the Synthesis and Biological Applications of Enoxacin-Based Compounds" by Suleiman et al. provides a systematic overview of the synthesis and evaluation of 4-quinolone-3-carboxylic acid derivatives. It specifically emphasizes scaffolds containing the 1,8-naphthyridine core, which is similar to enoxacin (ENX). The analysis addresses recent developments in the synthesis of these derivatives, highlighting their potential biological applications. "As classic precursor structures for antimicrobial agents, compounds derived from the 1,8-naphthyridine core exhibit activities such as antibacterial, antiviral, and antitumor effects. This paper provides a systematic summary of recent research progress in the field, which is significant for the further study and development of such compounds.
I recommend accepting the paper after the authors complete the revisions.
1. In Scheme 1, please indicate which elemental atoms are represented by 'X'. In the structure of compound 2, 'X' should be directly connected to the benzene ring.
2. In Scheme 6. Please list 17a-c as starting materials above the reaction arrow leading from ENX to 18a-c.
3. Line 215: “Compound 25” should be written as “Compound 24.”
4. In Table 6: “R” group for compounds 30c and 30d should be “O”, not”OH”.
5. In Table 6: the activity value of Compound 30a against E. coli should be '0.25' instead of '0-25'. Please correct all similar errors in the table.
6. In Table 8: the activity value of Compound 34c against E. coli should be '1.56', not '156”. Please carefully verify the activity data against the original literature."
Author Response
Reviewer 3
Comments and Suggestions for Authors
Comments:
The review manuscript titled "Advances in the Synthesis and Biological Applications of Enoxacin-Based Compounds" by Suleiman et al. provides a systematic overview of the synthesis and evaluation of 4-quinolone-3-carboxylic acid derivatives. It specifically emphasizes scaffolds containing the 1,8-naphthyridine core, which is similar to enoxacin (ENX). The analysis addresses recent developments in the synthesis of these derivatives, highlighting their potential biological applications. "As classic precursor structures for antimicrobial agents, compounds derived from the 1,8-naphthyridine core exhibit activities such as antibacterial, antiviral, and antitumor effects. This paper provides a systematic summary of recent research progress in the field, which is significant for the further study and development of such compounds.
I recommend accepting the paper after the authors complete the revisions.
Response:
We are grateful for your positive feedback and suggestions for improvement and have made the following changes:
- In Scheme 1, please indicate which elemental atoms are represented by 'X'. In the structure of compound 2, 'X' should be directly connected to the benzene ring.
Response1:
Thank you for your valuable feedback. The elemental atoms represented by X in Scheme 1 are X= O, S, NH. These elemental atoms are updated in scheme 1, page 4.
- In Scheme 6. Please list 17a-c as starting materials above the reaction arrow leading from ENX to 18a-c.
Response 2:
Thank you for your comment. 17a-c as previously denoted is now referred to as 18a-c. It is inserted as starting materials above the reaction arrow leading from ENX to 19a-c. This update can be found in page 8.
- Line 215: “Compound 25” should be written as “Compound 24.”
Response 3:
Thank you for your remark. The number of the compound has been correctly updated to 24. This correction can be found in Line 272, page 9, and on page 10 in Scheme 9.
- In Table 6: “R” group for compounds 30c and 30d should be “O”, not”OH”.
Response 4:
Thank you. The “R” group for compounds 30c and 30d in table 6 have been updated as “O” and can be found in table 6, page 11.
- In Table 6: the activity value of Compound 30a against E. coli should be '0.25' instead of '0-25'. Please correct all similar errors in the table.
Response 5:
Thank you. The activity value of compound 30a against E. coli on page 11, table 6 has been corrected as '0.25'. All similar cases in the manuscript have also been checked and corrected.
- In Table 8: the activity value of compound 34c against E. coli should be '1.56', not '156”. Please carefully verify the activity data against the original literature."
Response 6:
Thank you again for your remark. The activity value of compound 34c against E. coli in table 8, page 12, has been carefully verified and corrected based on the original literature source. Additionally, all similar cases in the manuscript have been reviewed and updated where necessary.

Round 2
Reviewer 2 Report
Comments and Suggestions for Authors
The main problem of this manuscript is not that it has potential overlapping phrases with ref. 62, Zahoor, et al., 2017, but that it actually duplicates this ref. Not in phrases, but in the content. The present review (Advances in the synthesis and biological applications of enoxacin based compounds) consists of three content parts: too long historical introduction, synthesis and biological activity. It is obvious that the synthetic part was reported earlier in the ref. 62, Zahoor, et al., 2017. In my opinion, synthetic part should be deleted from this manuscript. But in the present form, it is a clear reject.